# Hitting boundaries: Contract type, playing experience, non-cognitive skills, and sport anxiety in elite women cricketers

Kathryn Cross[1], Mark Daniel Batey[2]*, Andrew Denovan[3], Neil Dagnall[4], Daniel Powell[4]

1 England and Wales Cricket Board, Lord's Cricket Ground, London, United Kingdom, 2 Department of People & Performance, Manchester Metropolitan University Business School, Manchester, United Kingdom, 3 School of Psychology, Liverpool John Moores University, Liverpool, United Kingdom, 4 Department of Psychology, Manchester Metropolitan University, Manchester, United Kingdom

☯ These authors contributed equally to this work.
* m.batey@mmu.ac.uk

**Data Availability Statement:** Data cannot be shared publicly because of the small sample size and elite nature of the participants could lead to a loss of anonymity. Data are available from the

## Abstract

This paper examined the psychological impact of contextual influences (i.e., contract type and playing experience) on sport anxiety in elite women cricketers participating in The Hundred. A sample of 71 elite female cricketers playing during the 2021–2022 season took part. Forty-nine of the sample (69%) held professional contracts, and 22 (31%) had yet to sign a professional contract. Participants provided details about their contract type and playing experience and completed self-report measures assessing sport anxiety, mental toughness, and general self-efficacy. Since mental toughness and self-efficacy are non-cognitive constructs, which buffer competitive trait anxiety, analysis controlled for these variables. Multivariate analyses of covariance examined sport anxiety scores among participants in relation to Hundred matches played (either 0, 1–10, or more than 10) and contract type (whether participants had a professional contract in place or not). Subfactors of Worry, Somatic, and Confusion assessed sports anxiety. No significant main effects existed. However, alongside a significant interaction, a covariate mental toughness effect occurred. Examination of the interaction revealed Worry scores were lower in cricketers who were yet to play a Hundred match who had not received a professional contract. Furthermore, Worry and Somatic scores were higher in cricketers that had played more than 10 Hundred matches and had not received a professional contract. These findings have important implications for the development of elite women cricketers. Particularly, they highlight the need to differentially support players through their career progression.

Manchester Metropolitan University Research Data team (contact via researchdata@mmu.ac.uk) for researchers who meet the criteria for access to confidential data.

**Funding:** The author(s) received no specific funding for this work.

**Competing interests:** The authors have declared that no competing interests exist

# Introduction

## General context

This paper examined the psychological impact of contextual influences (i.e., contract type and playing experience) on sport anxiety in elite women cricketers participating in The Hundred cricket competition. This was a critical area of study because to date, despite the rapid growth of women's cricket, few studies have looked at these factors at the highest level of the women's game. The researchers focused on The Hundred because the tournament, launched in 2021, is a high-profile, well-financed competition. The importance of The Hundred stems from the fact that cricket authorities created the tournament to attract new spectators. Hence, the competition's launch and subsequent occurrences have used intense marketing campaigns to stimulate media and public interest [1]. Consequently, from the perspective of players, the tournament is imperative since involvement is fiscally rewarding and potentially career enhancing.

Although women's cricket has been an important British sport for decades [2], it has only recently transitioned from amateur (through semi-professional) to full professional status. The evolution of the limited over format illustrates this. Though the first Women's Cricket World Cup (50 overs aside) took place in 1973 and the inaugural T20 World Cup (20 overs aside) in 2009, the formation in England and Wales of the semi-professional Women's Cricket Super League (20 overs aside) did not take place until 2016. Indeed, progression from amateur to semi/full-time professional status has been slow and lagged behind the England women's cricket team successes. The England women won the Cricket World Cup in 1973, 1993, 2009, and 2017 (runners-up in 1978, 1982, 1988, and 2022) and triumphed in the T20 World Cup in 2009 (runners-up in 2012, 2014, and 2018).

A significant boost to the profile of the women's game was the launch of The Hundred in 2021 by the England and Wales Cricket Board (ECB). The competition was crucial because it afforded equal status to the women's and men's tournaments. Organisers achieved this by arranging fixtures in parallel, so that events are 'double-headers' with games occurring back-to-back at the same venue. This ensures large crowds for both women's and men's matches. In this context, for a substantial number of female cricketers, The Hundred provides the first consistent experience of playing in front of passionate crowds of thousands and television audiences of millions.

The spectating appeal of The Hundred is that it employs a unique format and team structure. In this context, the tournament is a very short form of the professional game with matches between two teams being composed of one innings each, lasting a maximum of 100 balls. This ensures that contests are intense and places an accent on fast, aggressive scoring. While the tournament borrows aspects from the established twenty over format (e.g., power plays where fielding restrictions are in place at the beginning of innings) it also possesses novel features such as bowlers delivering either 5 or 10 balls per ten ball end (a radical departure from conventional six ball overs) and strategic time-outs. In addition, to attract new audiences and provide wider appeal, loud DJ music plays during matches and live music between matches and changes of innings. Pyrotechnics accompany the scoring of a four or six (a boundary) or the taking of a wicket. Another distinct characteristic of The Hundred is that, rather than using the highly established regional County Cricket structure, it consists of specially created franchise teams located in major cities (i.e., Birmingham, London, Manchester, Leeds, Southampton, Nottingham, and Cardiff). For men's and women's fixtures both sets of cricketers play for the same franchise team against the same opponents.

Teams comprise squads of 15 players, recruited purely for The Hundred season. In the case of the men's tournament, teams populate squads using a draft system, whereas women's sides

use bespoke player selection. In both instances, a fixed, but generous budget restricts recruitment. The draft is a complex process where each team in turn selects players from a pool, which is composed of domestic and overseas players who have entered themselves. For each round there is a set value for player picks, starting at the highest and gradually reducing to a minimum value. The women's system differs because, despite having a long, established history, extant domestic squad depth is not conducive to the draft. For the first tournament, women's recruitment involved a two-stage open-market system with a standardised player fee band structure. In stage one, each team signed two current centrally contracted England Women's players. In the second phase, each team signed their remaining 13 players from across three different player pools: remaining England Women centrally contracted players, overseas players, and domestic players.

The ECB award central contracts to players who regularly represent their country. They are lucrative because they assure income commensurate with elite playing status. In the Women's game 2022–23, there were 18 centrally contracted players. Professional contracts for the women's game have rolled in since 2020 and increased steadily since due to enlarged funding. The desire to heighten playing standards and provide young women cricketers with a structured career pathway motivated this investment.

## The psychological importance of The Hundred

The Hundred is a crucial competition for women cricketers since the tournament provides an opportunity for them to display their talent alongside men, obtain commercial rewards, and enhance/consolidate status. Indeed, teams are likely to award lucrative contracts to high performing individuals, which assure player standing and provide economic stability. These factors, concurrent with the accelerating growth of the women's game, place differing psychological pressures on women cricketers as a function of contract type and competition experience [3].

Explicitly, the psychological pressures of performing for players without (vs. with) professional contracts are likely to vary as a consequence of the number of Hundred games played. Players with contracts should initially experience general anxiety regarding the need to perform to a standard commensurate with their status, which should then reduce as experience of The Hundred deepens. In comparison, levels of anxiety are likely to grow in players without professional contracts as the number of games increases. This is because, despite performing regularly at the elite level, they have yet to achieve a professional status. All of which occurs against a backdrop of intense media scrutiny and loud (sometimes boisterous) crowd environments.

These predictions are tentative since there is an absence of preceding psychological research in the domain of the women's elite game, and the cricketing environment and infrastructure are rapidly evolving. Nevertheless, research indicates that contract instability is a major source of anxiety in the men's game [4], and there are suggestions that the precarious nature of professional contracts in women's sports exacerbates anxiety. For example, elite women footballers report employment concerns related to absent childcare policies, low economic remuneration, short contracts, and limited post-career playing options [5]. Such factors are unique to elite women athletes, meaning their careers are potentially more uncertain and precarious than their male counterparts [6]. As such, elucidating the impact on anxiety of playing time with or without a professional contract in The Hundred could have important implications for elite women's sport.

## Non-cognitive skills: Mental toughness and self-efficacy

Player mindset (i.e., self-perceptions and beliefs) likely influences the interaction between contract type and playing experience. Acknowledging this, prior sport and performance-related research has reported that possession of mental toughness (MT) and self-efficacy (SE) moderate the negative effects of pressures and reduce undesirable affective states [7]. Theorists refer to these attributes and related constructs (resilience, hardiness, grit, etc.) as non-cognitive skills because they denote adaptive abilities not directly affected by intellectual capacity [8].

Mental toughness is an umbrella term for a range of psychological resources (e.g., the capacity to deal with adversity, self-belief, internal locus of control, resilience, persistence, and superior mental skills), which enable the ability to thrive under pressure, facilitate achievement, and promote positive mental health [9, 10]. Thus, MT cultivates positive values, attitudes, emotions, and thoughts. Although there are multiple definitions of MT, this paper draws on the conceptualisation of Clough et al. [11], who delineated MT as the ability to cope with difficulty and achieve self-defined goals. This delimitation is commensurate with the notion that MT protects against the adverse effects of stress. Within sport settings this occurs when environmental pressures evoke an appraisal process in which perceived demand exceeds resources producing unfavourable physiological, psychological, behavioural, or social outcomes [12]. In addition to sport, researchers have observed the benefits of MT across a range of real-world settings (i.e., occupational, health, and education).

Self-efficacy designates an individual's belief in their capability to attain desired goals. Consequently, like MT the construct is an important stress management resource [13, 14]. In a sporting context, SE denotes an athlete's belief that they can accomplish a task. Levels of SE derive from a combination of experience, personal qualities, and social support. Key to the development of SE is feedback, which allows athletes to appraise their progress and level of performance [15]. This information is important as it influences SE for subsequent learning and enactment (effort, persistence, and execution). While MT and SE are overlapping constructs that share common variance, they are psychometrically distinct [16]. Consistent with this delimitation, Nicholls et al. [17] postulated that MT sustains and/or enhances self-belief (efficacy) in challenging situations (i.e., when tasks are unfulfilling or stressful). This view is consistent with studies that report that MT is associated with strong faith in personal ability [11, 18].

Brace et al. [19] used the Goal-Expectancy-Self-Control (GES) model [20] as a theoretical framework for interpreting the relationship between MT and SE. The model developed from a review of MT research, which indicated that MT in stress producing situations was characterised by self-control, goals, and SE. The GES model is vital because as well as capturing crucial components of MT, the model explains how MT impacts on athletic performance. Noting this, and the fact that investigators have consistently reported that SE is related to sporting performance and success, Brace et al. [19] included SE in their study of elite ultra-marathon runners. The contribution of SE stems from the fact that perceived SE directs evaluations of effectiveness, particularly how well individuals believe they are able to execute actions needed to address prospective circumstances [13]. Central to this is the role of mastery, which facilitates persistence and the desire to sustain. Conversely, low SE may reduce application when individuals perceive tasks as too difficult to complete.

Within elite ultra-marathon runners, Brace et al. [19] found a strong positive correlation between MT and SE. A substantial proportion of shared variance was ascribable to Confidence (31%). This signified that a central common feature of MT and SE was self-belief (faith in personal ability and absence of doubts). However, MT and SE were not significantly related to actual performance (i.e., rank or likelihood of completing a 100-mile endurance run).

## Mental toughness and cricket

As stated previously, due to an absence of psychological research in the domain of elite women cricketers and the rapidly evolving nature of women's game, it is difficult to draw appropriate comparisons with previous research both within cricket and across women's elite sport. In the case of preceding work on cricket, research has focused on male cricketers [e.g., 21], although recent studies have begun to include female athletes [e.g., 22, 23]. The latter approach is more inclusive but limited since it assumes that playing conditions and pressures are equivalent. This is not necessarily the case since the women's game is rapidly advancing and there is less career stability due to restricted funding and the growing nature of professional infrastructure. These contextual factors also restrict the usefulness of generalisations between more established or emerging elite women's sports and cricket. In addition, the high profile, rapid, intense atmosphere of The Hundred competition makes for a distinct context in which to examine non-cognitive skills and sport anxiety. Nonetheless, it is important to consider previous academic work that has examined the role of MT in cricket since this is an established domain of psychological research, which affords insights into the psychological factors associated with playing elite cricket.

Pertinent to the present paper, studies have investigated the role that MT plays in the development of cricketers. For instance, Bull et al. [24] interviewed 12 male English cricketers identified by English Cricket Coaches as among the most mentally tough over the past two decades. They interpreted transcripts using a MT pyramid. This indicated that formative experiences such as upbringing (e.g., parental influence) and transition (seminal playing experiences) were important because they allowed players to encounter setbacks and learn from failure. Moreover, impact and range of experiences produced a foundation for tough character, attitudes, and thinking.

These outcomes aligned with subsequent research, which observed that experience played a significant role in the development of MT. For instance, Gucciardi [25] examining a sample of youth cricketers (aged between 13 and 18 years) reported the strongest predictors of MT were initiative experiences and negative peer influences. Initiative experiences (i.e., set goals, focus effort, plan, and problem solve) encouraged players to manage performance aspects effectively. Contrastingly negative peer influences (i.e., ridicule or pressure) reduced MT. Number of years playing and hours per week training failed to show strong relationships with MT. These findings suggested that it was type of experience rather than playing duration that was associated with higher levels of MT.

Within young cricketers, MT reflects possession of a set of enabling, constructive psychological attributes. Illustratively, Gucciardi et al. [26] found that young cricketers (aged between 10 and 18 years) with high (vs. moderate) levels of MT reported more internal (commitment to learning, positive values, social competencies, and positive identity) and external (support, empowerment, boundaries and expectations, and constructive use of time) developmental assets and lower levels of negative emotional states. Gucciardi et al. [26] concluded these characteristics, in addition to contributing to optimal performance, denoted thriving (i.e., the ability to succeed within the youth setting).

In addition to being associated with the development and success of young cricketers, studies have also found that MT is a crucial factor in adults. Particularly, Weissensteiner et al. [27] observed that highly (vs. lesser) skilled batters scored higher on MT. They postulated that this was because success required a combination of attributes embodied by MT. Specifically, the ability to cope with challenges, succeed, and persist despite demanding circumstances. Within the higher (vs. lesser) skilled batters, this expressed as greater self-belief, motivation, and higher levels of commitment and perseverance. Noting the importance of MT to elite

performance, Weissensteiner, et al. [27] recommended that sporting organisations should include the construct in talent identification and monitor MT within development programs.

The attributes identified by studies such as these are evident in the Cricket Mental Toughness Inventory (CMTI, [28]). The CMTI comprises five related dimensions affective intelligence (i.e., the capacity to manage arousal, anxiety, and stress), desire to achieve (i.e., determined to work hard and advance), resilience (i.e., capability to recover from setbacks and remain optimistic), attentional control (i.e., ability to stay focused and avoid distractions), and self-belief (i.e., self-confidence and absence of doubts).

### Present study

This paper examined the psychological impact of contextual influences (i.e., contract type and playing experience) on competition anxiety in elite women cricketers participating in The Hundred. Analysis controlled for the potentially mediating effects of MT and SE. These non-cognitive constructs are likely to influence outcomes because they are negatively associated with Competitive State Anxiety Inventory-2 (CSAI-2, [29]), highly positively correlated, and act as general stress buffers. Commensurate with this conceptualisation, previous studies have reported relationships between levels of non-cognitive skills, playing experience, and competitive anxiety in cricketers [24].

The association between MT and SE arises from the fact these constructs share common enabling factors. A recent study by Denovan et al. [see 16], who found that MT and SE loaded on a common Non-Cognitive Adaptive Resourcefulness dimension, demonstrated this. In this context, MT helps to sustain or enhance self-belief in stressful or unfulfilling activities [17]. This in turn, promotes belief in ability (i.e., competency to achieve desired goals) and SE.

Regarding, previous findings using the CSAI-2, this instrument is a state measure of cognitive anxiety, somatic anxiety, and self-confidence allied to sporting performance. The present study used the Sport Anxiety Scale (SAS, [30]) because it assesses competitive trait anxiety (i.e., somatic anxiety, worry and concentration disruption). The researchers selected trait anxiety because it is a more consistent, enduring measure of competitive concerns and uneasiness. Thus, higher trait anxiety tends to increase performance concerns and disrupt goal-related performance across a range of sporting contexts [31]. Cognizant of these factors the authors tested the following hypotheses:

H[1]: Mental toughness and self-efficacy will correlate negatively with sport anxiety.

H[2]: Cricketers with more playing experience would have lower sport anxiety (after controlling for mental toughness and self-efficacy), and

H[3]: Cricketers with a professional contract would have lower sport anxiety (after controlling for mental toughness and self-efficacy).

## Materials and methods

This study examined the psychological impact of contextual influences (i.e., contract type and playing experience) on sport anxiety in elite women cricketers participating in The Hundred cricket competition.

### Participants

The researchers purposively recruited participants. The sample comprised 71 elite female cricketers playing in The Hundred in England and Wales in the 2021–2022 season. This represented 59% of the total female players in the competition. Small sample sizes are common in elite sporting studies [e.g., 8, 19] given the restricted numbers of potential participants in the population. Twenty-eight players were aged between 18 and 23 years old (39%), 33 players

were aged between 24 and 29 years old (47%) and 10 players were aged over 30 (14%). To participate athletes had to be over the age of 18 and be a member of a Hundred women's squad. Excluded from participation were players under 18 years old who were not a member of a Hundred women's squad. Forty-nine of the sample (69%) held professional contracts, while 22 (31%) had yet to sign a professional contract. Regarding contracted players, their status was: 5 (10%) regional, 31 (63%) regional and Hundred, and Central 13 (27%). Players had held contracts: 28 (57,14%) 1–2 years, 11 (22.45%) 3–5 years, and 10 (20.41%). In terms of playing experience, 16 (22.5%) had previously played no Hundred games, 19 (26.8%) completed 1–10 Hundred games, and 36 (50.7%) more than 10 games. Data collection used categories to ensure that individuals players were not recognizable.

## Materials

Self-report measures assessed levels of Sports Anxiety, Mental Toughness, and General Self-Efficacy. Each scale used a five-point Likert scale ranging from (1 = Strongly Disagree to 5 = Strongly Agree) unless specified otherwise.

**The Sport Anxiety Scale-2 (SAS-2, [32]).** The Sport Anxiety Scale-2 (SAS-2) is an established instrument that assesses levels of anxiety pre and during competition/performance. The scale consists of 15 items that represent 3 (5 item) subscales: Worry (i.e., concerns about performing poorly and ensuing negative consequences), Somatic Anxiety (i.e., autonomic arousal centred in the stomach and muscles), and Concentration Disruption (i.e., difficulties focusing on task-relevant cues). Instructions outline the fact that athletes often feel tense or experience nervousness before or during competitions and games. Participants then worked through the items (i.e., 'I feel nervous') indicating level how they usually feel during or before Hundred competition matches. A 4-point Likert type scale (1-not at all to 4-very much) collected responses. The SAS-2 has attested psychometric properties. Specifically, acceptable internal consistency at both total score and subscale levels, test-retest reliability, and validity [32].

**Mental Toughness Questionnaire, 10-item (MTQ-10, [33, 34]).** The researchers assessed mental toughness using the 10-item Mental Toughness Questionnaire (MTQ-10, [33, 34]). The instrument is a shortened, psychometrically enhanced version of the 18-item Mental Toughness Questionnaire (MTQ-18, [11]), which has featured prominently in published research. The MTQ-18 is a unidimensional scale derived from the multidimensional 48-item Mental Toughness Questionnaire (MTQ-48, for a recent review see [10]). The MTQ-18 contains the high loading items from each of the four dimensions of the MTQ-48 (Challenge, Commitment, Control, and Confidence). Accordingly, investigators have widely adopted the MTQ-18 even though the measure lacks full psychometrically evaluated. Acknowledging this, Dagnall et al. [34] scrutinised the MTQ-18. Examination revealed that although the scale was an adequate instrument. supplementary variance arising from MTQ-48 multidimensionality contaminated MTQ-18 structure. Addressing this issue, produced the psychometrically superior MTQ-10.

Researchers developed abridged versions of the MTQ-48 because large test batteries due to length restrictions require a succinct, general MT measure. Thus, the MTQ-10 allows researchers to assess MT alongside a range of constructs without placing an undue cognitive burden on participants. Items within MTQ measures appear as statements (e.g., "I generally feel that I am a worthwhile person"). Totalling of MTQ-10 produces a global score. The MTQ-10 possesses sound psychometric properties (i.e., internal reliability, validity, and invariance) [34].

**General Self-Efficacy Short Scale (GSE-S, [35]).** The General Self-Efficacy Short Scale (GSE-S) is a 3-item instrument that evaluates self-belief in competence. Explicitly, ability to perform, across a range of achievement situations. The GSE-S derived from the established

10-item scale measure developed by Schwarzer et al. [36]. Since the developers created the GSE-S for use within surveys it focuses on brevity. This was crucial in the context of the present sample to ensure engagement and maximise survey completions by busy athletes. The GSE-S has demonstrated satisfactory psychometric properties across national samples (i.e., reliability, validity, and invariance) [35].

## Procedure

Following institutional ethical approval (Manchester Metropolitan University ETHoS Review #47690) the lead investigator contacted each athlete via email, WhatsApp or through the Professional Cricketers' Association (PCA). Instructions outlined the study and directed potential participants to the online survey. Only athletes who provided informed consent and met the inclusion criteria advanced to the self-report measures. Each athlete explicitly provided informed consent. Alongside rating scales participants completed a demographics section. This comprised: age, whether English was the athlete's first language, contract status, playing role in the team and number of The Hundred fixtures played. To limit common method variance, instructions encouraged psychological separation by stressing differences between constructs [37]. Data collection occurred between 21$^{st}$ September and 1$^{st}$ November 2022.

## Data analysis strategy

Following initial examination of data (i.e., screening, consideration of descriptive statistics, and performance of zero-order correlations) three multivariate analyses of covariance (MANCOVA) were utilised to investigate Sport Anxiety scores among cricket players in relation to the Number of The Hundred Matches Played (either 0, 1–10, or more than 10) and Contract Type (yet to sign professional contract vs. professional contract in place). To examine Anxiety independent of the influence of Mental Toughness and Self-Efficacy, analysis entered these variables as covariates. Each MANCOVA included No. of One Hundred Matches Played and Contract Type as independent variables. Somatic Anxiety was the dependent variable (DV) in the first MANCOVA, Worry in the second, and Confusion in the third. Univariate analyses and post-hoc pairwise comparisons followed up significant main effects.

## Results

### Descriptive statistics

Descriptive statistics appear in Table 1, revealing the highest category of Anxiety was Worry among cricket players who had played more than 10 The Hundred matches, and were yet to receive a professional contract. The lowest Anxiety category was Confusion among cricket players who had played between 1–10 The Hundred matches and were yet to sign a professional contract.

### Inferential statistics

Initial examination of the relationships between continuous variables Anxiety and Mental Toughness (MT) ($r = -.64$, $n = 71$, $p < .001$), and Anxiety and Self-Efficacy (SE) ($r = -.20$, $n = 71$, $p = .045$) revealed significant negative associations (Table 2). MT and SE exhibited a significant positive correlation ($r = .38$, $n = 71$, $p < .001$). Gignac and Szodorai [38] concluded (based on meta-analyses) that correlations of .10, .20, and .30 represented small, typical, and large correlations. Using these guidelines, the associations within this study were typical to large. To control for the potential effects of MT and SE on dependent variables, analysis included these constructs as covariates within the MANCOVA.

**Table 1. Mean and standard deviation of No. of Hundred matches played, contract type, and anxiety type [a].**

| No. of Hundred Matches Played | Contract Type | Anxiety Type | | |
| --- | --- | --- | --- | --- |
| | | Somatic | Worry | Confusion |
| 0 | Yet to sign professional contract | 10.0 ± 3.21 | 13.07 ± 3.70 | 9.38 ± 3.25 |
| 1–10 | Yet to sign professional contract | 10.60 ± 1.51 | 15.0 ± 3.60 | 8.20 ± 1.64 |
| More than 10 | Yet to sign professional contract | 14.50 ± 4.04 | 17.5 ± 1.29 | 9.25 ± 2.36 |
| 0 | Professional contract in place | 12.0 ± 3.60 | 15.0 ± 4.0 | 8.33 ± 1.52 |
| 1–10 | Professional contract in place | 11.07 ± 3.97 | 14.71 ± 3.31 | 8.92 ± 2.97 |
| More than 10 | Professional contract in place | 9.90 ± 3.18 | 13.31 ± 3.64 | 9.37 ± 3.53 |

[a] Data presented as *Mean ± SD*

Initial inspection of MANCOVA assumptions revealed a non-significant result for Box's Test of Equality of Covariance Matrices ($p = .319$), indicating that homogeneity of variance-covariance matrices existed. Moreover, a non-significant Levene's result occurred for each dependent variable (Somatic $p = .204$, Worry $p = .299$, Confusion $p = .700$). These results indicated satisfaction of basic assumption.

MANCOVA revealed a non-significant main effect of Contract Type, Wilks' λ = .99, $F(3, 61) = .12$, $p = .951$, $\eta^2 = .01$ (small effect), and Number of Hundred Matches, Wilks' λ = .94, $F(6, 122) = .59$, $p = .735$, $\eta^2 = .03$ (small effect). A significant interaction existed, Wilks' λ = .74, $F(6, 122) = 3.32$, $p = .005$, $\eta^2 = .14$ (medium effect). Moreover, a significant effect of MT (as a covariate) occurred, Wilks' λ = .52, $F(3, 61) = 18.30$, $p < .001$, $\eta^2 = .47$ (large effect). Univariate analyses revealed, for the interaction, a significant effect in relation to Worry, $F(2, 63) = 5.59$, $p = .006$, $\eta^2 = .15$ (large effect) and Somatic, $F(2, 63) = 5.71$, $p = .005$, $\eta2 = .15$ (large effect).

Pairwise post-hoc comparisons, using a Bonferroni correction (rescaled to significance $p < .05$ by SPSS), revealed that Worry scores were significantly lower in cricketers who were yet to play a Hundred match who had not received a professional contract (vs. cricketers who had a professional contract) ($p = .027$). Furthermore, Worry ($p = .018$) and Somatic ($p = .010$) scores were significantly higher in cricketers that had played more than 10 matches who had not received a professional contract (vs. cricketers who had a professional contract). Figs 1 and 2 illustrate interaction relationships. Accordingly, analysis demonstrated that Worry and Somatic scores differed in relation to contract type and number of Hundred matches (see Table 1).

## Discussion

Consideration of zero-order correlations revealed that MT was positively correlated with SE and negatively correlated with Sports Anxiety. These observed relationships were large [see 38]

**Table 2. Associations among continuous study variables.**

| Variable | M | SD | 1 | 2 | 3 | 4 | 5 | 6 |
| --- | --- | --- | --- | --- | --- | --- | --- | --- |
| 1. MTQ-10 | 3.34 | .53 | | .38** | -.64** | -.53** | -.64** | -.41** |
| 2. GSE-S | 3.63 | .52 | | | -.20* | -.17 | -.23* | -.10 |
| 3. SAS-2 | 2.24 | .56 | | | | .83** | .89** | .77** |
| 4. Somatic | 2.11 | .68 | | | | | .64** | .41** |
| 5. Worry | 2.79 | .72 | | | | | | .55** |
| 6. Confusion | 1.83 | .62 | | | | | | |

*Note.* MTQ-10 = 10-item Mental Toughness Questionnaire; GSE-S = General Self-Efficacy Short Scale; SAS-2 = Sport Anxiety Scale-2; *$p < .05$; **$p < .001$

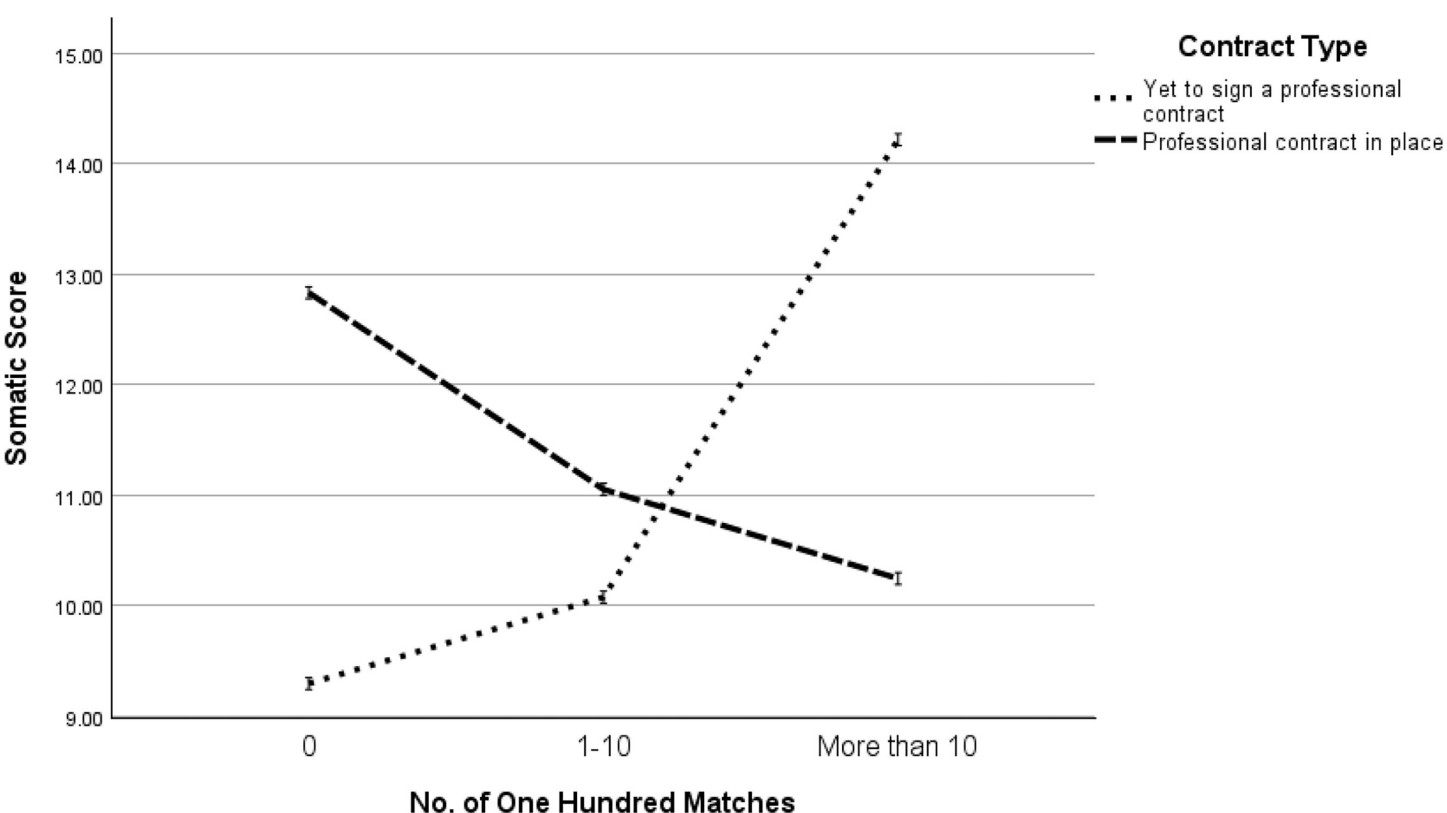

Fig 1. Interaction of No. of The Hundred matches and contract type in relation to Somatic scores.

and consistent with the conceptualisation of MT as a set of psychological resources that assist the capacity to cope with performance-related pressure [9, 10]. In the context of The Hundred, relationships indicated that MT related to self-belief in competence (Self-Efficacy) and the capacity to manage Worry (i.e., concerns about performing poorly and subsequent negative consequences), Somatic Anxiety (i.e., autonomic arousal), and Concentration Disruption (i.e., difficulties focusing on task-relevant cues). These attributes aligned well with dimensions of the Cricket Mental Toughness Inventory (CMTI, [28]). Particularly, affective intelligence attentional control (i.e., ability to stay focused and avoid distractions) and self-belief (i.e., confidence in ability and absence of doubts).

Despite being positively correlated with MT, SE only negatively correlated with Worry. This outcome was consistent with the operationalisation of SE as faith in personal ability and lack of doubts [19]. The absence of significant relationships with Somatic and Confusion indicated that SE, as measured by the GSE-S, is a narrower construct than MT, which comprises elements of Challenge, Commitment, Control, and Confidence. Since the relationships were correlational, it was not possible to determine the extent to which the observed relationships were directional and/or interactive. Therefore, results require cautious interpretation.

Examining differences on non-cognitive skills as a function of Contract Type, players with Professional (vs. No Professional) contract scored higher on MT, there was no difference for SE. The difference reflects the fact that contextual factors (i.e., positive playing/career-related occurrences) enhance elements of MT. This conclusion is consistent with prior research, which reports that significant foundational experiences shape the development of MT in

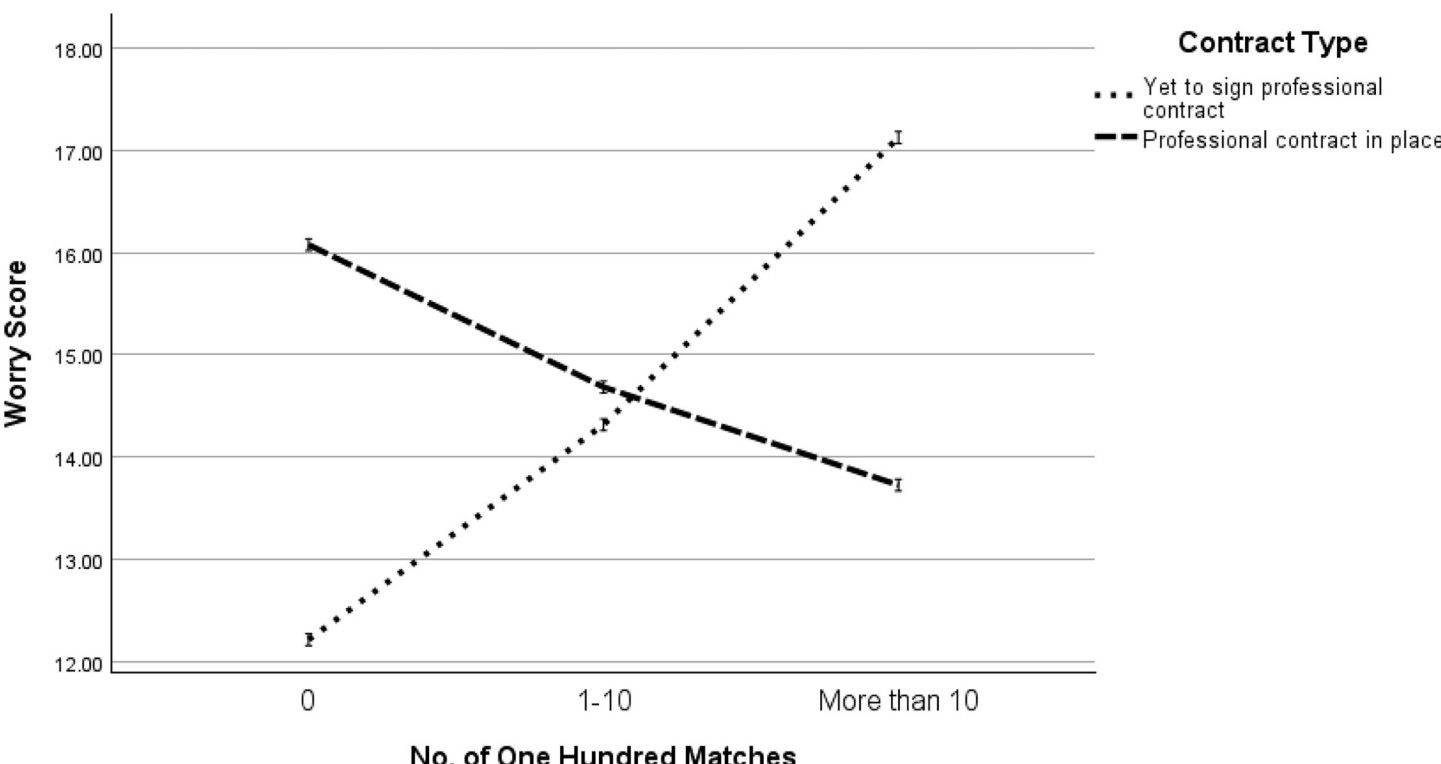

Fig 2. Interaction of No. of The Hundred matches and contract type in relation to Worry scores.

cricketers [24, 25]. Hence, the reassurance of a contract represents an important initiative experience, which enables players to better set career goals, focus effort, and plan. Furthermore, a contract affords a sense of security, whereby players are able to focus on development and have latitude to experiment, and even fail. In the case of players with professional contracts, the need to retain or obtain an enhanced contract also produces psychological and performance-related pressures. Future studies should also investigate these in the context of the women's elite game.

The outcomes from this study have important implications for the development of women's elite cricket. Particularly, they highlight the need to support players through their progression from amateur, though semi-professional, to full-time professional status. This is especially true with regards to Hundred players who have yet to attain a contract. Support could include psychological training to deal with the pressures of playing at an elite without career stability and mentoring so that emerging players are able to draw on the experiences of established professionals and centrally contracted peers. These types of enabling provision will help players deal with the demands of elite cricket.

This is crucial because this study found that performing in the Hundred without a professional contract resulted in higher levels of tournament-related competitive anxiety (Worry and Somatic; concerns about performing poorly and allied negative consequences). Although, sports-related anxiety can be both facilitative and debilitative to performance, over a sustained period its influence is typically negative. Explicitly, sport-related anxiety can adversely impact performance (i.e., training, practice, and competitions) and injury susceptibility (i.e., risk,

recovery, and reinjury) [39]. This is especially possible when players continue to participate, or teams retain them without progressing to a professional contract.

Whilst this issue is presently most likely to affect players aspiring to gain professional contracts, it will subsequently impact upon professionals. As the number of professional players increases and competition for squad places intensifies then professional players will increasing face contract competition and uncertainty. Accordingly, this is something that subsequent research should assess. Nonetheless, the present study indicated that both at national body and tournament level, players would benefit from career support structures. This is difficult in the context of The Hundred because it is a franchise tournament, where players join teams solely for the purpose of the tournament then return to their regional County Cricket clubs.

Additionally, players may benefit also from non-cognitive skills training. Preceding investigations have reported that increased knowledge and awareness of intrapersonal (e.g., attitudes, learning strategies, drives and self-regulation) and interpersonal (relationships with others) domains related to performance can assist performance, enhance MT and sense of SE, and improve the capacity to cope with career challenges [7]. Hence, knowledge of non-cognitive skills will help players with elevated levels of anxiety and who are experiencing career uncertainties. Specifically, support work within the professional body and club infrastructure will prepare players for the demands of transitioning, obtaining/maintaining professional contracts, and performing at elite level. Acknowledging this, MT is crucial since it promotes a positive mental mindset, which encourages solution-focused coping and protects against the negative effects of stress. These attributes are concomitantly conducive to an individual's belief in their capability to attain desired goals (i.e., level of self-efficacy) [16].

Although being exploratory, the current investigation provided important insights into the effects of contract type and playing experience on performance-related anxiety in elite women cricketers playing in The Hundred. This research was necessary because of the absence of psychological research in women's cricket generally and specifically as the sport is currently transitioning to full-time professional status. This shift places pressures on players to obtain and maintain full-time contracts and perform at the highest levels. This is especially true in The Hundred due to the competition's high media profile and the lucrative financial rewards, which are potentially career enhancing. Thus, elite women cricketers presently represent an under-researched population. Accordingly, identifying factors that cause players performance-related anxiety is important as this can also affect their sense of well-being and development. In this context, if the women's game is to continue to grow and mature it is vital that the cricketing infrastructure supports player progress. Only then, will the sport have the level of domestic squad depth required to ensure consistently high, sustainable levels of performance. That is not to say that the current standards are not excellent but recognizes concerns regarding the size of the current professional playing pool.

Although the findings of this study are important, researchers should view outcomes with caution due to limitations. One important concern is sample size. As outlined in the introduction, the pool of elite women (vs. men) cricketers, despite increased funding and expansion of the game remains small. Consequently, the population available to sample is restricted. Illustratively, there are currently eight teams that compete in the two premier competitions organised by the England and Wales Cricket Board, the Charlotte Edwards Cup (20 over) and the Rachael Heyhoe Flint Trophy (50 over competition).

Similarly, in the case of The Hundred there are eight teams with squads of 15 players (n = 120 players). This means that any study of elite domestic women's cricketers, even with high rate of engagement, will only be able to recruit small numbers compared to general studies. The main concerns with small samples are underpowered analysis and increased error margins increase. These issues are problematic because they can result in under-representative

outcomes that do not correctly reflect the relationships/differences under observation. Consequently, the use of small samples can undermine validity and limit generalisability. The current sample (n = 71) was good because it represented a high percentage of women players who had participated in The Hundred. This is especially true since at the time of data collection, the competition had only run for two years. Hence, the majority of players participated in the inaugural competition and clubs retained them for the second season.

The number of participants was also relatively high given the typical operational (training and playing) and organisational (clubs manage players time and duties) demands women Hundred players face, therefore are a difficult population to access. Thus, numbers in the present study were commensurate with those in previously published investigations using specialist and/or elite sporting populations. For instance, Weissensteiner et al. [27] examined differences in adult-aged batters of two different skill levels (high, *n* = 11 vs. lesser, *n* = 10) on a battery of psychological tests including MT, perfectionism, coping ability, and optimism. Bell et al. [40], in their study 2-year longitudinal intervention examining whether mental toughness enhanced performance under pressure in elite young cricketers, tested 41 (*n* = 20 intervention vs, *n* = 21 control) male cricketers aged between 16 and 18. Similarly, non-cognitive skills studies with specialist populations have also drawn on small, convenience samples (e.g., international rugby league players, *n* = 70, [41]; English Premier League academy football players, *n* = 112, [42]; English academy football players, *n* = 73, [43]; professional Welsh football team players, *n* = 20, [44]; and players in the Football Association Women's Super League, n = 63, [8]).

Although small samples are problematic statistically, examples illustrate how they are a necessity when working with highly-specialist populations such as elite sportspersons. In this context, responses provide crucial insights into the self-reported cognitions and perceptions of elite women cricketers. Ensuing studies should extend the present study and seek to replicate the reported findings.

Regarding measurement of MT, this study used the unidimensional MTQ-10 [33, 34]. While this is a psychometrically-sound scale, it provides only a global score. The current study employed the MTQ-10 because its' brevity was likely to facilitate participation and completion of survey measures. Long test batteries are time consuming and subsequently, elite players struggle to fit them in alongside intense training and playing schedules. Indeed, outside of club and playing commitments they have restricted availability to engage with research. This was why this investigation employed short, easy to complete measures (i.e., MTQ-10, GSE-S, and SAS-2). Though this expedient approach facilitated a sizeable number of responses and completions, scores on the measures provided only snapshots of complex constructs. Noting this, subsequent investigations should assess mental toughness using the multidimensional, MTQ-48. This will provide more nuanced insights into relationships mental toughness, contract type, playing experience, and competition anxiety. Explicitly, indicate whether there are differential effects as a consequence of Challenge, Commitment, Control, and Confidence.

Another limitation of the present study was that it employed a self-report, cross sectional design, where the investigators collected data at only one point in time. This approach is problematic as results could reflect a temporal artefact (i.e., reflect a restricted time period). As stated previously, this is a concern due to the rapid development of the women's elite game in England and Wales. Acknowledging this, future studies should use a longitudinal design that examines relationships across multiple time points over an extended period. This would establish the stability of the effects and provide causal insights that could conceptually inform the development of a predictive model. Additionally, to enhance measurement peer ratings would enable the assessment of key constructs [8].

Finally, to ensure that the results are not context specific, proceeding work should examine the impact of career uncertainties such as contract security on performance-related anxiety in

other settings such as Academies, ECB tournaments, and other international competitions (e.g., The Women's Indian Premier League). This will indicate whether The Hundred, due to its high profile and potential rewards places greater pressures on women cricketers.

## Conclusion

The present paper demonstrated that contract type and playing experience affected levels of sport anxiety in elite women cricketers participating in The Hundred. Particularly, cricketers without a professional contract who had not previously played were less Worried (i.e., concerned about performing poorly and ensuing negative consequences). This was probably due to lower perceived expectations. In contrast, professional players new to the format feel greater pressure to achieve.

Additionally, Worry and Somatic anxiety (i.e., autonomic arousal centred in the stomach and muscles) were higher in cricketers that had played more than 10 matches who had not received a professional contract. This likely reflected concerns about subsequent playing status and career. This complex interaction demonstrated that contract type and playing experience had a significant impact on performance concerns. In the case of professional cricketers their status created initial pressures, which eases as the number of games increased. Whereas for players without a professional contract performance concerns and arousal increased as a function of matches played.

A further important outcome was the association between mental toughness and lower levels of sports anxiety. Consistent with previous literature, this suggested that mental toughness facilitated self-efficacy and enabled individuals to mitigate performance-related anxiety. However, this supposition requires caution since the observed relationships were correlational and non-casual. Nonetheless, this paper has important implications for the development of elite women cricketers. Particularly, it highlights the need to differentially support players through their development and career progression and indicates that they would benefit from non-cognitive skills training.

## Author Contributions

**Conceptualization:** Kathryn Cross, Mark Daniel Batey.

**Data curation:** Kathryn Cross, Andrew Denovan.

**Investigation:** Mark Daniel Batey, Andrew Denovan.

**Methodology:** Mark Daniel Batey.

**Project administration:** Kathryn Cross, Andrew Denovan, Neil Dagnall, Daniel Powell.

**Supervision:** Mark Daniel Batey.

**Writing – original draft:** Kathryn Cross, Mark Daniel Batey.

**Writing – review & editing:** Mark Daniel Batey, Andrew Denovan, Neil Dagnall, Daniel Powell.

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
