## [Decision Letter · Decision Letter 0]

14 Feb 2024

PONE-D-23-38171Hitting Boundaries: Contract type, playing experience, non-cognitive skills, and sport anxiety in elite women cricketersPLOS ONE

Dear Dr. Batey,

Thank you for submitting your manuscript to PLOS ONE. After careful consideration, we feel that it has merit but does not fully meet PLOS ONE’s publication criteria as it currently stands. Therefore, we invite you to submit a revised version of the manuscript that addresses the points raised during the review process.

**Thank you for submitting your manuscript to our journal, PLOS ONE. Although your manuscript meets the publication criteria of this journal, one of the reviewers has pointed out some shortcomings. Kindly revise your manuscript accordingly and resubmit. **

We look forward to receiving your revised manuscript.

Kind regards,

Uzair Yaqoob

Academic Editor

PLOS ONE

Journal Requirements:

2. You have indicated that data is available from [researchdata@mmu.ac.uk].  Please can we ask you to provide us with a general contact email address for the data requests, so readers can request access in perpetuity. If a general email is not available please provide a link to a website where readers can obtain access to data. 

Reviewers' comments:

Reviewer's Responses to Questions

**Comments to the Author**

1. Is the manuscript technically sound, and do the data support the conclusions?

Reviewer #1: Yes

Reviewer #2: Partly

2. Has the statistical analysis been performed appropriately and rigorously? 

Reviewer #1: Yes

Reviewer #2: Yes

3. Have the authors made all data underlying the findings in their manuscript fully available?

Reviewer #1: Yes

Reviewer #2: No

4. Is the manuscript presented in an intelligible fashion and written in standard English?

Reviewer #1: Yes

Reviewer #2: No

5. Review Comments to the Author

Reviewer #1: The authors of the study have determined correlation among several cognitive and non-cognitive measures such as mental toughness, self-efficacy, nature of contract, and the previous playing experience regarding the overall sporting performance among a group of 71 elite cricketers. This is an interesting study, highlighting the factors and ways forward in which sport anxiety and worry can be minimized to enhance professional performance. As much as we know that the lack of foundational experiences such as temporary nature of the contract and job insecurity does cause worries and anxieties in the performers, especially if they are on the field, performing live in front of thousands of spectators, the results are nonetheless redundant. However, the study has a decent statistical analysis, and the results could be published as an interesting read to reiterate the need of supporting young female cricketers and providing them with the psychological support.

Reviewer #2: The subject of sports anxiety in female cricketers has yet to receive comprehensive study, and as such, the authors are to be commended for selecting this area of research. However, various modifications are necessary to refine the current study.

Sample size was not calculated. There are few grammatical errors. Revisions are required.

Major Revisions:

Page 12 line 278: Do not mention the analysis information here. You can mention the study setting, sampling technique used, and sample size calculation here

Page 13 line 282: How was the sample size determined? Did any other study have a similar number of participants? Please mention the reason. Also, mention the sampling technique used.

Page 13 line 283: Write the inclusion and exclusion criteria properly. In inclusion criteria mention the age, professional or not, contract status, and experience. Then explain the exclusion criteria. Also, exclude the statistics and mention them in the results part.

Page 16 line 348-58: Add this section at the end of methodology.

Page 16 line 359: this should be added at the start of results section.

Page 16 line 360-63: Mention the direction and strength of the correlation.

Page 16 line 363: Kindly provide the context of this reference. Also write the context in discussion section.

Page 16 line 367: enter the percentages from page 13 line 281 here as demographics.

Page 17 line 370: Add descriptive statistics first and then continue with the inferential statistics.

Page 24: Add conclusion

Minor Revisions:

Page 2 line 42-3: write sub-factors in inverted commas for clarity

Page 7 line 155 and 162: Do not repeat words like mental toughness and self-efficacy. Use acronyms. Also do mention it in the brackets first. Repetition of words has been consistently noted.

Page 8 line 174: write the acronym in brackets for Goal-Expectancy-Self-Control model.

Page 12 line 261-62: please explain this point with respect to the given reference.

Page 17 line 390: mention the Hundred in inverted commas for clarity.

Page 17 line 390-92: Use punctuation marks and correct the sentence structure.

Page 21 line 467: correction needed: *against

6. PLOS authors have the option to publish the peer review history of their article (what does this mean?). If published, this will include your full peer review and any attached files.

Reviewer #1: No

Reviewer #2: **Yes**

---

## [Decision Letter · Decision Letter 1]

22 Jul 2024

Hitting Boundaries: Contract type, playing experience, non-cognitive skills, and sport anxiety in elite women cricketers

PONE-D-23-38171R1

Dear Dr. Batey,

We’re pleased to inform you that your manuscript has been judged scientifically suitable for publication and will be formally accepted for publication once it meets all outstanding technical requirements.

Kind regards,

Uzair Yaqoob

Academic Editor

PLOS ONE

Additional Editor Comments (optional):

Reviewers' comments:

Reviewer's Responses to Questions

**Comments to the Author**

1. If the authors have adequately addressed your comments raised in a previous round of review and you feel that this manuscript is now acceptable for publication, you may indicate that here to bypass the “Comments to the Author” section, enter your conflict of interest statement in the “Confidential to Editor” section, and submit your "Accept" recommendation.

Reviewer #2: (No Response)

2. Is the manuscript technically sound, and do the data support the conclusions?

Reviewer #2: (No Response)

3. Has the statistical analysis been performed appropriately and rigorously? 

Reviewer #2: (No Response)

4. Have the authors made all data underlying the findings in their manuscript fully available?

Reviewer #2: (No Response)

5. Is the manuscript presented in an intelligible fashion and written in standard English?

Reviewer #2: (No Response)

6. Review Comments to the Author

Reviewer #2: (No Response)

7. PLOS authors have the option to publish the peer review history of their article (what does this mean?). If published, this will include your full peer review and any attached files.

Reviewer #2: **Yes: **Tabeer Tanwir Awan

---

## [Editor Report · Acceptance letter]

1 Aug 2024

PONE-D-23-38171R1 

PLOS ONE

Dear Dr. Batey, 

I'm pleased to inform you that your manuscript has been deemed suitable for publication in PLOS ONE. Congratulations! Your manuscript is now being handed over to our production team.

Kind regards, 

on behalf of

Dr. Uzair Yaqoob 

Academic Editor

PLOS ONE